# Efficiency of Coagulation/Flocculation for the Removal of Complex Mixture of Textile Fibers from Water

Sanja Vasiljević [1], Maja Vujić [1], Jasmina Agbaba [1], Stefania Federici [2,*], Serena Ducoli [2], Radivoj Tomić [1] and Aleksandra Tubić [1,*]

1 Department of Chemistry, Biochemistry and Environmental Protection, Faculty of Sciences, University of Novi Sad, Trg Dositeja Obradovića 3, 21000 Novi Sad, Serbia

2 Department of Mechanical and Industrial Engineering, University of Brescia and INSTM Unit of Brescia, via Branze, 38, 25123 Brescia, Italy

* Correspondence: stefania.federici@unibs.it (S.F.); aleksandra.tubic@dh.uns.ac.rs (A.T.)

**Abstract:** Synthetic fibers enter wastewater treatment plants together with natural fibers, which may affect treatment efficiency, a fact not considered in previous studies. Therefore, the aim of the present study was to evaluate the efficiency of the coagulation/flocculation process for the removal of a mixture of textile fibers from different water matrices. Natural and synthetic fibers (100 mg/L; cotton, polyacrylonitrile, and polyamide) were added to a synthetic matrix, surface water and laundry wastewater and subjected to coagulation/flocculation experiments with ferric chloride ($FeCl_3$) and polyaluminum chloride (PACl) under laboratory conditions. In the synthetic matrix, both coagulants were found to be effective, with $FeCl_3$ having a lesser advantage, removing textile fibers almost completely from the water (up to 99% at a concentration of 3.94 mM). In surface water, all dosages had approximately similar high values, with the coagulant resulting in complete removal. In laundry effluent, the presence of surfactants is thought to affect coagulation efficiency. PACl was found to be effective in removing textile fibers from laundry wastewater, with the lowest removal efficiency being 89% and all dosages having similar removal efficiencies. Natural organic matter and bicarbonates showed a positive effect on the efficiency of $FeCl_3$ in removing textile fibers from surface water. PACl showed better performance in coagulating laundry wastewater while surfactants had a negative effect on $FeCl_3$ coagulation efficiency.

**Keywords:** textile fibers; coagulation and flocculation; wastewater; removal of textile fibers; ferric chloride; polyaluminium chloride

## 1. Introduction

In recent years, the presence of microplastic fibers (MPFs) in wastewater attracted considerable concern and attention, as synthetic fibers became one of the most commonly used materials in the manufacture of clothing, along with natural fibers [1]. Every year, more than 42 million tons of synthetic fibers are produced in the textile industry [2–4], most of which end up in the environment. In the environment, MPFs are mostly recognized as small fibers released during the production and use of textiles [5–7]. Polyester and nylon are the most common polymers used for the production of textile fibers, accounting for about 63% of the total textile materials, but also polyamide, polystyrene, polyacrylonitrile, etc. are frequently used [7]. Based on previous research, it was proved that the formation of MPFs is influenced by the type of detergent, the type of washing machine, and the program temperature, among other factors. It is assumed that parameters such as high temperature and water hardness further promote the formation of MPFs, while the addition of softeners reduces their number [8]. MPFs enter the environment through two main pathways: laundry waste that directly enters the environment or municipal wastewater in wastewater treatment plants [9–11].

Different types of microplastics (e.g., polyester, polyethylene, polypropylene, polyamide, etc.) enter wastewater treatment plants in different forms (flakes, fibers, films, and particles) and from different sources (from synthetic textile washing processes and microbeads from household products, such as cosmetics, detergents, eyeglass cleaners, etc.) [12–14]. It was found that 6 million polyester fibers and 700,000 acrylic fibers are released from one washing of textiles in the machine, which depends on the washing program, detergent, and textile properties [12]. Previous research indicated that synthetic MPF should be given special attention because they are very small and narrow and can be retained very easily in wastewater [15].

Advanced technologies for the removal of microplastics from water such as biodegradation, adsorption, catalysis, photocatalytic degradation, coagulation, filtration, and electrocoagulation are being investigated to some extent but are still considered as under-researched technologies, especially in wastewater treatment plants [16]. Coagulation and flocculation technology is considered as a typical treatment for the removal of suspended solids and colloidal particles in wastewater. Researchers focused on this treatment in recent years, mainly because of its low cost, ease of operation, and energy savings [17]. Several factors affect the efficiency of removal of synthetic fibers by coagulation/flocculation, such as pH, natural organic matter, surfactants, ions present in the matrix, coagulation conditions, etc.

The pH is a very important factor in coagulation treatment. At lower pH values, the main mechanism of coagulation is believed to be neutralization of charge between positively charged Fe/Al hydroxypolymers and negatively charged particles. The removal of particles at higher pH values is mainly due to the adsorption of particles on Fe/Al precipitates. Previous studies showed that PACl is effective at slightly acidic pH, but also at higher pH values, which depends on the properties of the matrix [18].

In surface waters, there is a large amount of natural organic matter (NOM) consisting of humic substances containing humic and fulvic acids characterized by a large molar mass. In addition, they are rich in aromatic and phenolic structures, conjugated double bonds, and hydroxyl and carboxyl groups. Therefore, NOM can change the surface properties of the fibers and thus affect the treatment [17].

The presence of surfactants in water can affect the efficiency of coagulation/flocculation treatment by altering polymer surface properties and affecting binding to coagulants. It was found that nonionic surfactants in particular can reduce coagulation efficiency [19,20].

In previous research, coagulation in wastewater treatment plants was used sparingly to remove microplastics. Most of the research in this area focused on studying the treatment of municipal wastewater, since wastewater treatment plant effluents are considered the main source of pollution of microplastics in the environment [4,8,21,22]. Available literature provides data on the coagulation/flocculation efficiency in removing different types of microplastic particles (polyethylene, polyester, polystyrene, rayon, etc.) with different sizes and shapes, untreated or aged. Common coagulants, such as iron and aluminum salts were used in the experiments. For the study of the coagulation/flocculation mechanism, synthetic matrices are usually prepared under laboratory conditions with well-defined properties in terms of pH, content of natural organic matter, metal salts, etc. [23–25]. However, the obtained results vary depending on the coagulation conditions, polymer types and properties, and matrix characteristics. Therefore, the need for studies related to more realistic matrices and different types of synthetic fibers was pointed out [7,26].

Based on these considerations, the aim of this work was to investigate the potential of coagulation/flocculation treatment in the removal of a complex mixture of textile fibers from natural and synthetic materials. Two common coagulants (ferric chloride and polyaluminum chloride) were used, as well as the potential of their combination. Wastewater from textile washing and surface water were selected as representatives of real water matrices, and the results were compared with those obtained for a synthetic matrix. In this way, we were able to evaluate the influence of natural organic matter, surfactants, and pH on the efficiency of removing complex textile fibers from real water matrices by coagulation/flocculation treatment.

## 2. Materials and Methods

### 2.1. Experimental Setup

In the present study, the efficiency of removing complex mixtures of textile fibers from water by coagulation and flocculation was investigated. Three different water matrices (synthetic water, surface water, and laundry wastewater) were used to evaluate the effects of different matrix properties. The removal of textile fibers was studied by adding 50 mg of textile fibers to the studied water matrices (0.5 L). The efficiency of the commonly used coagulants ferric chloride (FeCl$_3$) and polyaluminum chloride (PACl) and their combination was evaluated. The experiments were performed under laboratory conditions using the standard JAR test. Figure 1 shows a schematic diagram of the experiment.

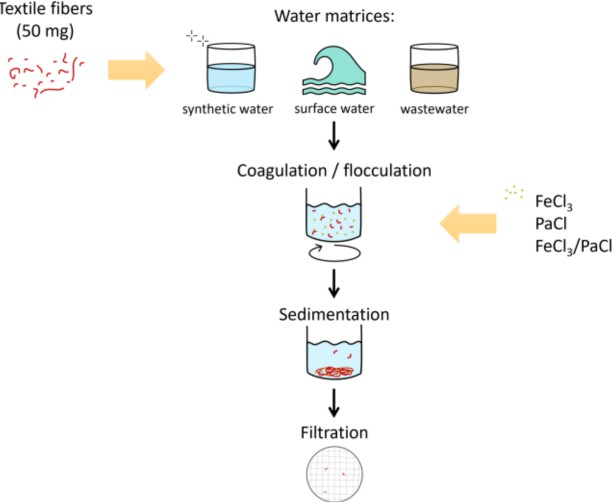

**Figure 1.** Schematic representation of the experiment.

### 2.2. Materials

Three waters were studied to have representatives of different matrix properties: laundry wastewater, surface water from the Danube River, synthetic matrix (distilled water enriched with NaHCO$_3$, CaCl$_2$, and MgSO$_4$ salts), with the properties presented in Table 1.

Mixed fibers containing cotton and synthetic polymers were used as material. A commercial detergent was used to approximate the actual washing conditions as closely as possible. A "mixed" cycle at 40 °C for 1 h and 40 min with a spin speed of 800 rpm was used as the wash program. The wastewater was collected at the outlet pipe of the machine in polyethylene terephthalate canisters. Both detergent and fabric softener were used to wash the textile garments, and the laundry was rinsed four times. It was then dried in a dryer, and the residue on the filter after the laundry was dried and used as material for further experiments. The concentration of textile fibers was 100 mg/L.

To compare the process of coagulation and flocculation with and without added textile fibers, samples without textile fibers (blank tests) were prepared.

FeCl$_3$ stock solution was prepared by dissolving 10 g of solid FeCl$_3$ (obtained from Sigma Aldrich, USA) in 50 mL of distilled water and used for dosing in water samples. The PACl stock solution was prepared by dissolving 7.7 mL of a PACl stock solution in 100 mL of distilled water. The dosage of coagulants is listed in Table S1. The stock solution of the flocculant (UNIFLOC M27, Unichem KFT, Hungary) was prepared by dissolving 0.5 g of the powdered substance in 100 mL of distilled water with stirring at 10 rpm in a JAR apparatus for 120 min. A working solution was prepared from the stock solution of the flocculant (c = 0.05 mL/mL) by dissolving 2.5 mL of the stock solution in 50 mL of distilled water so that the concentration in the experiment was 0.025 mL/mL.

**Table 1.** Characteristics of the water matrices.

| Parameter | MDL | Synthetic Water | Surface Water | Laundry Wastewater |
|---|---|---|---|---|
| pH | / | $7.90 \pm 0.06$ | $7.75 \pm 0.07$ | $8.13 \pm 0.07$ |
| Electrical conductivity 25 °C (μS/cm) | / | $303 \pm 5$ | $437 \pm 8$ | $578 \pm 15$ |
| Turbidity (NTU) | 0.01 | $0.3 \pm 0.3$ | $8.35 \pm 0.15$ | $149 \pm 25$ |
| COD (mg/L) | 15 | <15 * | <15 * | $920 \pm 23$ |
| Total organic carbon (mg/L) | 0.3 | <0.3 * | $2.4 \pm 0.3$ | $104 \pm 1.1$ |
| $Na^+$ (mg/L) | 0.1 | $75.7 \pm 5.7$ | $5.5 \pm 0.7$ | $2.71 \pm 0.22$ |
| $Mg^{2+}$ (mg/L) | 0.09 | $4.33 \pm 0.41$ | $10.4 \pm 0.9$ | $5.01 \pm 0.47$ |
| $K^+$ (mg/L) | 0.11 | <0.11 * | $2.31 \pm 0.17$ | $6.8 \pm 0.6$ |
| $Ca^{2+}$ (mg/L) | 0.11 | $44.0 \pm 3.6$ | $6.00 \pm 0.54$ | $7.33 \pm 0.71$ |
| $Cl^-$ (mg/L) | 5.0 | $52.1 \pm 3.6$ | $44.0 \pm 1.5$ | $3140 \pm 150$ |
| $SO_4^{2-}$ (mg/L) | 5.0 | $21.2 \pm 4.9$ | $25.5 \pm 3.2$ | $97.6 \pm 27.2$ |
| $HCO_3^-$ (mg/L) | 18.4 | $134 \pm 6$ | $218 \pm 43$ | $252 \pm 32$ |

* method detection limit (MDL).

## 2.3. Coagulation/Flocculation Experiments

The coagulation experiments were performed on the JAR apparatus under laboratory conditions. All experiments were performed in duplicate or triplicate. Before performing the JAR test, 50 mg of textile fibers were added to 0.5 L of water and allowed to stand for 24 h before the coagulation/flocculation experiments. Two common coagulants (ferric chloride and polyaluminum chloride) were used. Coagulation was performed by adding the coagulant at a mixing speed of 120 rpm for a period of 2 min, after which a flocculant was added and mixing was continued at 45 rpm for a period of 30 min. Sedimentation was performed for 30 min. After settling, the samples were filtered through a cellulose nitrate membrane filter with a pore size of 0.45 μm using a vacuum filtration device.

The dosage of coagulant followed the dosages commonly used in the literature [26] and was adjusted according to water requirements (Table S1). Therefore, lower dosages (0.72–3.95 mM Fe and 0.77–3.85 mM Al) were used in the synthetic matrix and surface water experiments, while higher dosages (7.16–21.49 mM Fe and 1.92–9.62 mM Al) were required for coagulation of laundry wastewater. The coagulant concentrations for laundry wastewater were higher due to the surfactants and other substances present in this water.

In addition, two coagulants were combined in that the dosage of $FeCl_3$ and PACl was lower than that which showed the best performance in removing textile fibers when used separately. Three dosing approaches were carried out: (1) $FeCl_3$ was added first and then PACl, (2) PACl was added first and then $FeCl_3$, and (3) the coagulants were added simultaneously.

The filter papers were dried in Petri dishes at room temperature for 24 h, and then their masses were measured, based on which further calculations were performed and the efficiency of removing textile fibers from water was determined.

## 2.4. Analytical Procedures

Textile fibers were characterized using a scanning electron microscope SEM (TM3030, Hitachi, Japan) to study their size, surface area, and shape. Chemical characterization was performed using Fourier transform infrared spectroscopy (FTIR), specifically the attenuated total reflection (ATR) technique on a FTIR Nexus 670 (Thermonicolet, USA)instrument in the range 4000–400 $cm^{-1}$ with a resolution of 4 $cm^{-1}$ and a speed of 60 scans per analysis at room temperature. The ATR technique was implemented using an ATR crystal of Ge on a reflective plate and a pressure clamp. In addition, characterization was also performed using micro-FTIR. FTIR measurements were performed using a Nicolet iN10MX (Thermo

Scientific, Waltham, MA, USA) microscope equipped with a cooled MCT detector and operating in transmission mode. For each sample, 20 individual fibers were isolated and placed on a $BaF_2$ window to collect IR spectra in transmission mode. Measurements were made with a nominal spectral resolution of 8 cm$^{-1}$ in the range of 4000–650 cm$^{-1}$. To increase the signal-to-noise ratio, 64 scans per sample were co-added without changing the position of the sample between each scan (total acquisition time: 12 s, including dead time). OMNIC$^{TM}$ Picta software (Thermo Scientific, Waltham, MA, USA) was used for all spectra manipulations.

The pH of the water samples was measured using a pH meter 340i, WTW, SenTix$^{®}$ 21 electrodes. Electrical conductivity analysis was performed using a Hanna conductometer, model HI 933000, Austria. Total organic carbon (TOC) in water samples was analyzed using an LiquiTOCII, Elementar, Germany instrument. Turbidity was determined using a WTW Austria, Cond 3210 with a Tetracon 325 WTW electrode. The chemical oxygen demand (COD) was determined according to the standard method [27].

Chloride concentration in water was determined by titration with silver nitrate using a potassium chromate indicator according to the method SRPS ISO 9297/1:2007 [28]. Sulfate was determined by iodometric titration of the excess chromate ion with sodium thiosulfate solution after precipitation of sulfate with the addition of excess barium chromate. The alkalinity of water in terms of bicarbonate and carbonate concentration was determined by the volumetric method [29].

Sodium and potassium metal ions were detected using the flame atomic emission spectroscopy (FAES) technique according to the standard method [29], and Ca and Mg metal ions were detected using the flame atomic absorption spectroscopy (FAAS) technique on an Perkin Elmer Analyst 700 atomic absorption spectrometer, according to the USEPA methods [30,31].

## 3. Results

### 3.1. Water Matrix Characterization

The characteristics of the water matrices studied are shown in Table 1. All three waters have similar pH values in the slightly alkaline range (pH = 7.75 ± 0.07–8.13 ± 0.07) to evaluate the efficiency of the coagulation/flocculation process under the conditions of the real pH of the water matrices.

The obtained results show that, as expected, laundry wastewater has the highest organic matter load, with a TOC content about 50 times higher than in surface waters and a COD of 920 ± 23 mg/L. The synthetic matrix is characterized by TOC and COD values lower than the detection limit of the method. This is followed by a high turbidity of the effluent and a very low (<1 NTU) for the other two matrices. As expected, the electrical conductivity shows the highest ion content in laundry wastewater, followed by surface water and synthetic matrix. When evaluating the influence on textile fiber behavior during coagulation/flocculation treatment, pH, TOC, and COD are the most important water quality parameters, as are $Cl^-$, $SO_4^{2-}$, and $HCO_3^-$ [25,26].

### 3.2. Textile Fibers Characterization

3.2.1. SEM Characterization

The different appearance of textile fibers in SEM micrographs indicates the presence of fibers of both natural and synthetic origin. A detailed visual characterization was performed using SEM and the results are shown in Figure 2. SEM microscopic images showed differences in the diameter of isolated textile fibers, ranging from 8.28 μm to 19.00 μm. Moreover, the different fiber structure in the samples is visible on the images obtained by SEM, indicating irregular, uneven, and heterogeneous fiber shapes. Fibers with a heterogeneous shape are usually of natural organic origin, such as cotton and wool, while fibers with a smooth structure are of synthetic, typically polymeric, origin [32]. A wide diameter range also indicates presence of different types of fibers in the selected sample [33].

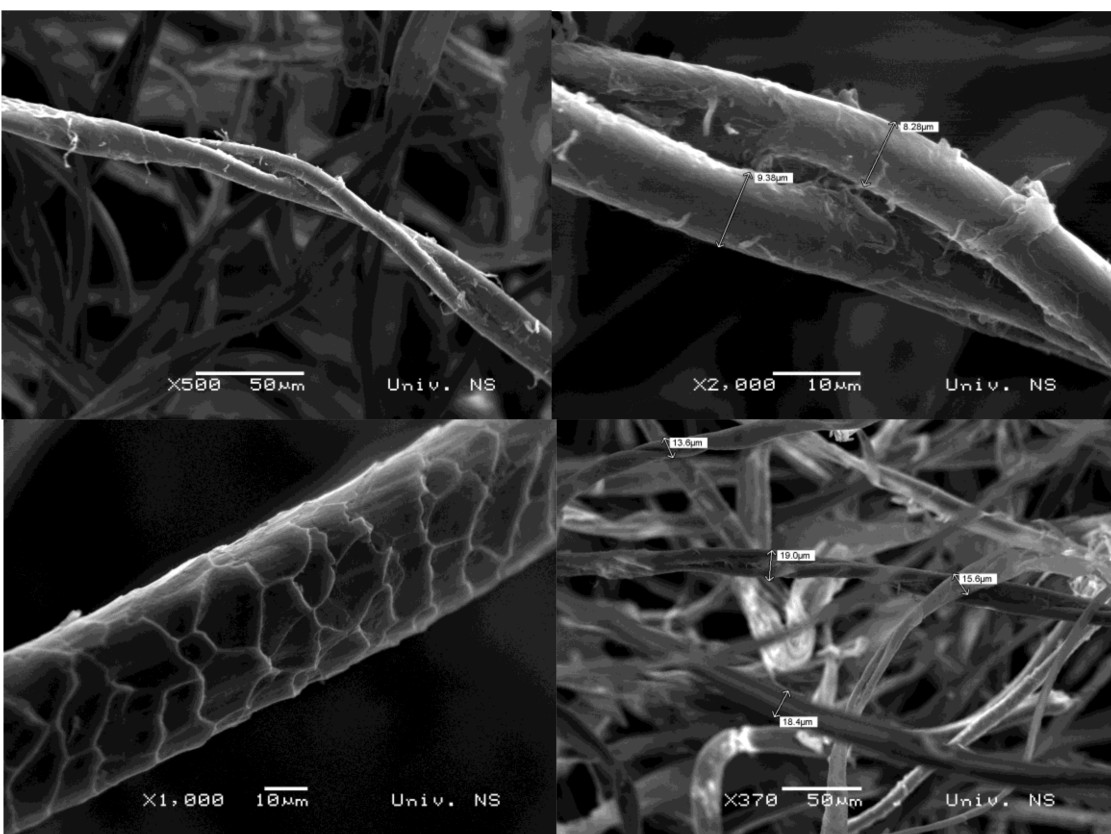

**Figure 2.** SEM microscopic images of textile fibers.

The SEM results of also indicate that there is a wide range of fiber diameters that can enter the water matrix, which means that this is one of the parameters that can affect the efficiency of removal processes, such as coagulation/flocculation [33].

### 3.2.2. FTIR

To determine the functional groups present in the textile fibers, the FTIR spectra were recorded in the range of 4000–400 cm$^{-1}$ (Figure 3). Based on the FTIR technical library "Synthetic fibers under the microscope", 51% of the isolated fiber sample analyzed was identified with rayon. Considering that the specific rayon/viscose peaks are 2917.4 cm$^{-1}$ associated with an aldehyde functional group (-C-H, weak bond), 1093.9 cm$^{-1}$ aliphatic amines (C-N stretch, strong bond) and 1017 cm$^{-1}$ alkyl fluoride (C-F stretch, strong bond), it is possible that other types of textile fibers were also part of the selected sample [34].

The main differences between the spectral data of the microscopic library and the obtained spectrum of the synthetic fibers are 1467, 1198, and 1155 cm$^{-1}$ in viscose. Moreover, the FTIR spectra of the analyzed isolated sample indicate the presence of a large amount of organic components in the analyzed sample. The presence of organic components, presumably derived from cotton, in the analyzed samples can be confirmed by the specific vibrations at the wavelengths 3250–4000 cm$^{-1}$, as suggested by Geminiani et al. (2022) [35].

### 3.2.3. Micro-FTIR

To further improve the characterization of the chemical composition of the textile fibers, additional analyses were performed using micro-FTIR. Furthermore, sample purification was performed according to the methods described in the literature [36–38]. Figures 4 and 5 show representative FTIR spectra and images of textile fibers.

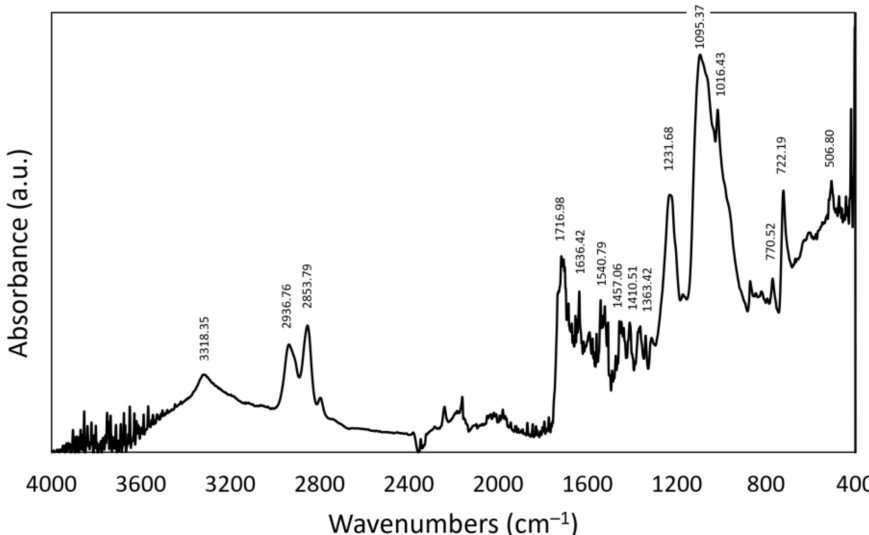

**Figure 3.** FTIR spectrum of textile fibers.

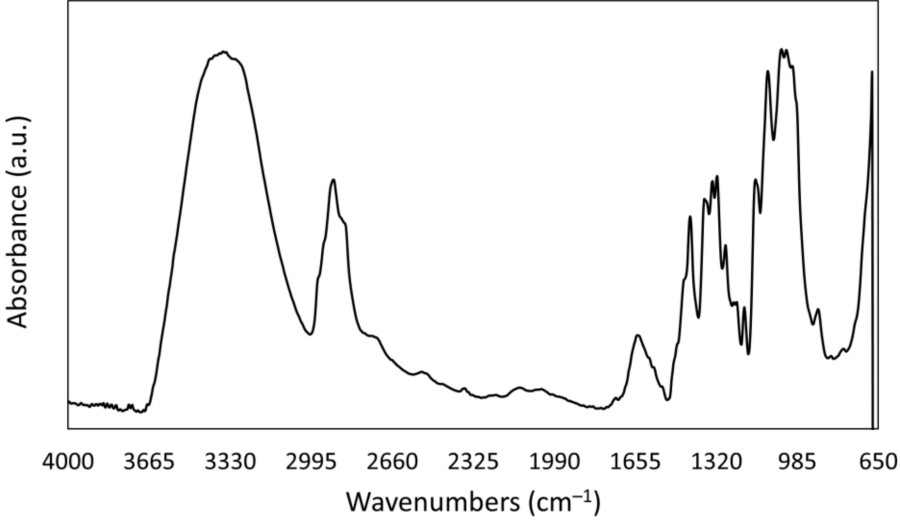

**Figure 4.** Representative FTIR spectrum of complex textile fibers (a total of 20 randomly selected fibers were analyzed).

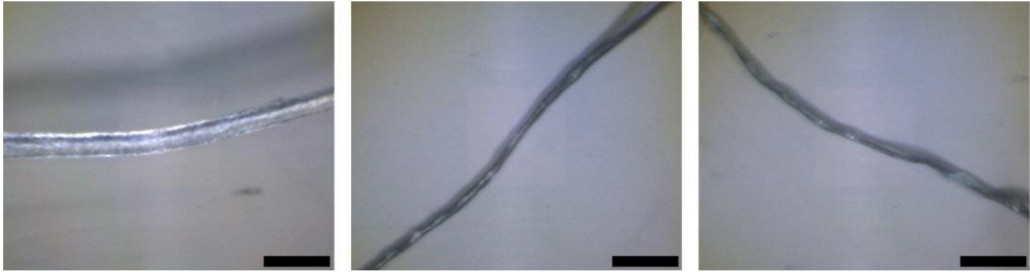

**Figure 5.** Images of a representative fiber mixture analyzed with an FTIR spectrometer.

The spectrum shown in Figure 4 shows characteristic absorption bands matching to the characteristics of cotton fibers: peak at 3360 cm$^{-1}$ corresponding to intramolecular and intermolecular hydrogen bond (O-H) stretching vibrations; peaks at 2900 cm$^{-1}$ and 2860 cm$^{-1}$ corresponding to asymmetric and symmetric -CH$_2$ stretching vibrations; peak at 1740 cm$^{-1}$ corresponding to carbonyl group (C=O) stretching vibrations; peak at 1650 cm$^{-1}$ corresponding to O-H bending vibrations of adsorbed water; peak at 1433 cm$^{-1}$

corresponding to -CH$_2$ in-plane bending vibrations; peaks at 1370 cm$^{-1}$, 1339 cm$^{-1}$, and 1322 cm$^{-1}$ corresponding to C-H bending vibrations, O-H in-plane bending vibrations, and C-H out-of-plane bending vibrations, respectively; peak at 1284 cm$^{-1}$ corresponding to C-H deformation stretching vibrations; peak at 1208 cm$^{-1}$ corresponding to O-H in-plane bending vibrations; peaks at 1160 cm$^{-1}$ and 1109 cm$^{-1}$ corresponding to asymmetrical bridge C-O-C stretching vibrations; peaks at 1054 cm$^{-1}$, 1034 cm$^{-1}$, and 1008 cm$^{-1}$ corresponding to C-O stretching vibrations; peak at 905 cm$^{-1}$ corresponding to asymmetric out-of-plane ring stretching vibrations and β-glucoside bond vibrations [39,40]. All 20 fibers analyzed in the sample had the same spectral profile, indicating that natural fibers (cotton) predominate and synthetic fibers are difficult to identify.

Due to the limited ability to distinguish synthetic fibers in the mixture of textile fibers, sample purification and re-characterization was required. The sample was purified with 100 mL of 30% hydrogen peroxide and dried for 24h, after which it was re-characterized (Figures 6 and 7).

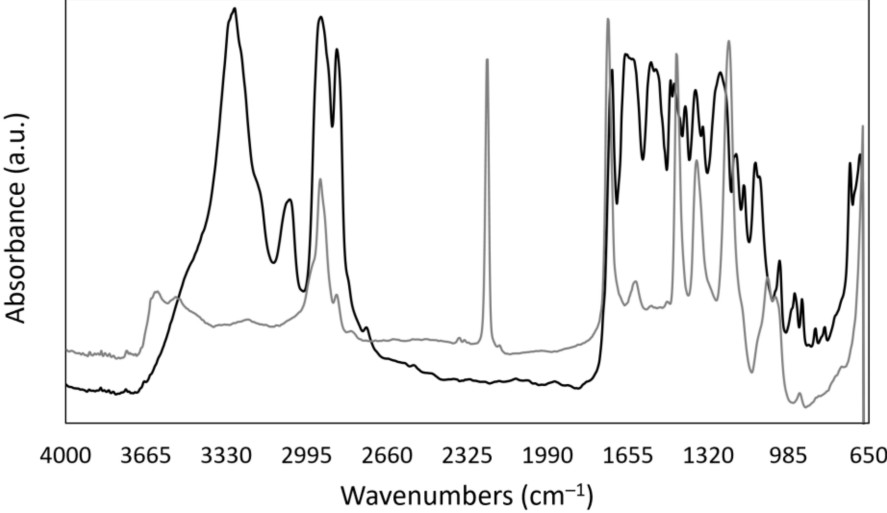

**Figure 6.** Representative FTIR spectrum of synthetic fibers (black spectrum: polyamide; grey spectrum: polyacrylonitrile). A total of 20 randomly selected fibers from a purified sample were analyzed.

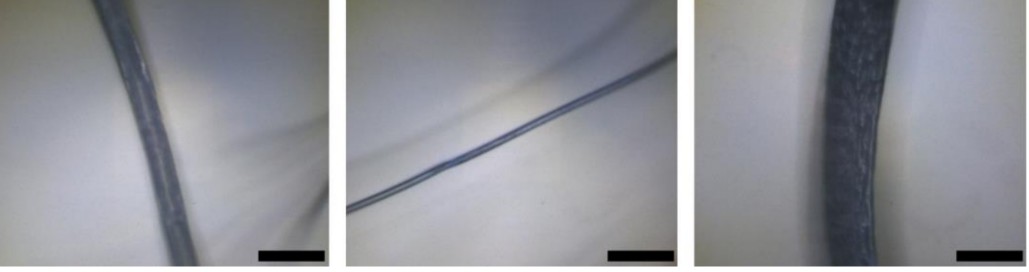

**Figure 7.** Images of representative synthetic fibers analyzed by FTIR spectrometer.

The black spectrum in Figure 6 shows characteristic absorption bands corresponding to the features of polyamide polymer: peak at 3300 cm$^{-1}$ corresponding to N-H stretching vibrations; peak at 3080 cm$^{-1}$ corresponding to N-H bending vibrations; peaks at 2931 cm$^{-1}$ and 2861 cm$^{-1}$ corresponding to stretching vibrations of methylene groups (-CH$_2$); peak at 1635 cm$^{-1}$ corresponding to stretching vibrations of carbonyl groups (C=O) (amide I); peak at 1535 cm$^{-1}$ corresponding to N-H bending vibrations and C-N stretching vibrations (amide II); peak at 1370 cm$^{-1}$ corresponding to C-N stretching vibrations; peak at 1141 cm$^{-1}$ corresponding to out of plane bending vibrations of carbonyl (C=O) groups; and peaks at 933 cm$^{-1}$ and 906 cm$^{-1}$ corresponding to N-H stretching vibrations [41–43].

The grey spectrum presented in Figure 7 shows characteristic absorption bands matching the characteristics of the polyacrylonitrile polymer: namely, peaks at 3626 cm$^{-1}$ and 3545 cm$^{-1}$ corresponding to asymmetric and symmetric O-H stretching vibrations, respectively; peaks at 2944 cm$^{-1}$ and 2875 cm$^{-1}$ corresponding to asymmetric and symmetric -CH$_2$ stretching vibrations; peak at 2245 cm$^{-1}$ correlating with C≡N stretching vibrations; peaks at 1740 cm$^{-1}$ and 1630 cm$^{-1}$ corresponding to C=O stretching vibrations (attributed to a small amount of vinyl acetate monomers in the polyacrylonitrile structure); peaks at 1458 cm$^{-1}$ and 1375 cm$^{-1}$ corresponding to -CH$_2$ and C-H bending vibrations; peak at 1240 cm$^{-1}$ corresponding to C-O stretching vibrations (vinyl acetate); and peak at 1074 cm$^{-1}$ corresponding to -CH$_2$ stretching vibrations [44,45].

Of the 20 randomly analyzed fibers in purified textile fibers, the spectra match corresponded to about 20% polyamide and about 80% polyacrylonitrile, reflecting the composition of the original synthetic compounds. A fiber with a high match to polyester was also discovered. This type of fiber is one of the most frequently documented sources of microplastic contamination in textiles, which consist of about 60% synthetic fibers [46].

### 3.3. Effects of Coagulation on Textile Fibers Removal Efficiency

To evaluate the efficiency of coagulation/flocculation for removing complex textile fiber mixtures from water, different dosages of coagulants were tested. The optimal dosage for the selected coagulants was determined by comparing the removal efficiency.

Figure 8 and Table S2 show the removal efficiency as a function of coagulant concentration in the synthetic matrix.

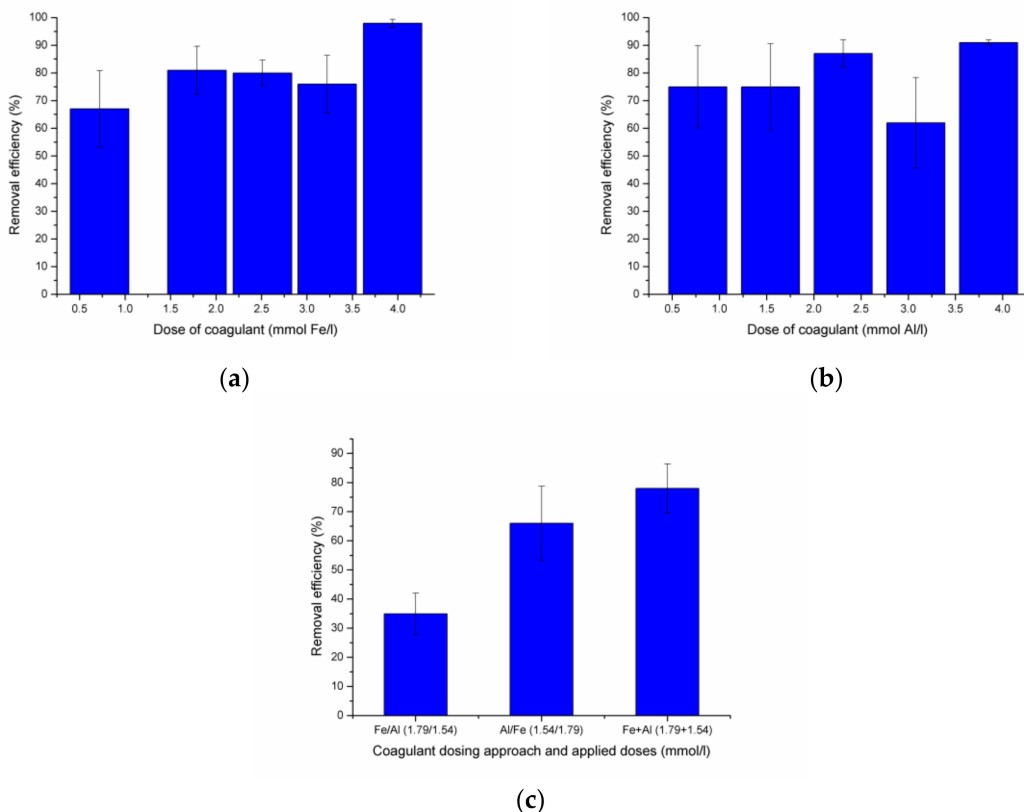

**Figure 8.** Efficiency of removal of textile fibers from a synthetic matrix by coagulation with (**a**) FeCl$_3$, (**b**) PACl, and (**c**) a combination of FeCl$_3$ and PACl (error bars show standard deviation of duplicates).

From the results shown in Figure 8, coagulation and flocculation with the application of $FeCl_3$ in the synthetic matrix 67–99% removal of textile fibers can be achieved, compared to the initial concentration. The highest removal efficiency was achieved at a $FeCl_3$ concentration of 3.94 mM, while the lowest was at a $FeCl_3$ concentration of 0.72 mM. It can be observed that all applied doses showed similar values for removal efficiency, except for the highest dose, indicating a complete removal. The use of PACl as a coagulant allowed a slightly lower removal efficiency with values in the range of 57–91% compared to coagulation with $FeCl_3$. The removal efficiencies for the combination of coagulants in this treatment were 35%, 66%, and 78% for Fe/Al, Al/Fe, and Fe+Al dosages, respectively. These values indicate that the combination of coagulants has no effect on improving the removal efficiency of textile fibers compared to single coagulants.

In the results obtained with both coagulants, it can be noted that when a dose of 3.22 mM is applied, the efficiency of coagulation begins to decrease significantly. This can be explained by the fact that the micro flakes of the coagulant tend to loosen and break when a large amount of coagulant is used, reducing the performance of the process, which was also confirmed by Zhou et al. 2021 [25].

Figure 9 and Table S2 show the efficiency of the removal of textile fibers from surface water using two separate coagulants and their combination.

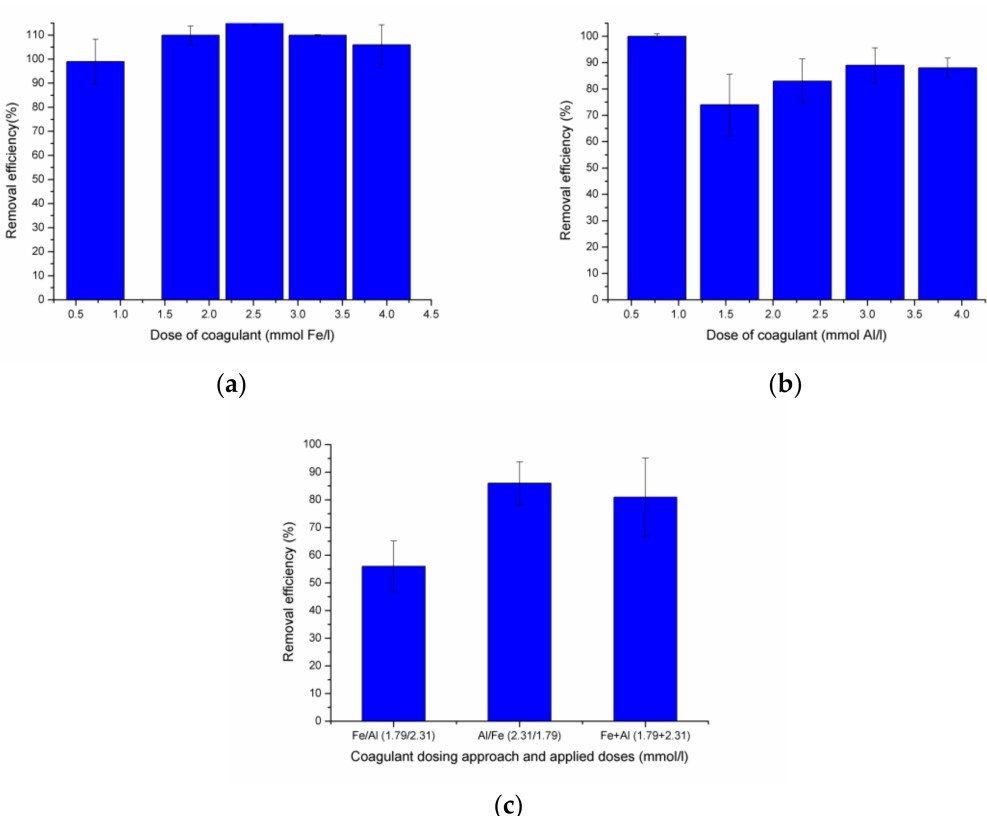

**Figure 9.** Efficiency of the removal of textile fibers from a surface water by coagulation with (**a**) $FeCl_3$, (**b**) PACl, and (**c**) a combination of $FeCl_3$ and PACl (error bars show standard deviation of duplicates).

Removal of textile fibers from surface water with $FeCl_3$ proved to be effective, as shown by the values in the plot (Figure 9a). All dosages had approximately similary high values, with the coagulant resulting in complete removal. Coagulation with $FeCl_3$ resulted in additional precipitation of particles from the water, as in this case the calculated amounts of material removed were higher than the amounts of textile fibers added to the matrix. The removal of textile fibers with PACl also proved to be an effective treatment, as shown by the values of more than 74% compared to the initial concentrations. The combination of $FeCl_3$ and PACl did not significantly improve the removal of textile fibers compared to the use of

the individual coagulants. The removal values for the combination of these two coagulants were 56%, 86%, and 81% for the dosages of Fe/Al, Al/Fe, and Fe+Al, respectively. For both the two coagulants used and their combination, the removal efficiency of textile fibers is higher than that of the synthetic matrix (Figure 8). It can be assumed that the natural organic matrix and the $HCO_3^-$ present in natural water contribute to a higher removal efficiency compared to the results obtained in synthetic matrix, which is also pointed out by other authors [25].

Coagulation and flocculation treatment of laundry wastewater required higher dosages of coagulants compared to synthetic matrix and surface water due to the presence of surfactants in the water (Table 1). Lower dosages were also investigated, but since no precipitation occurred, the results are not presented. The coagulation/flocculation efficiency of textile fibers in wastewater treatment is shown in Figure 10 and Table S3.

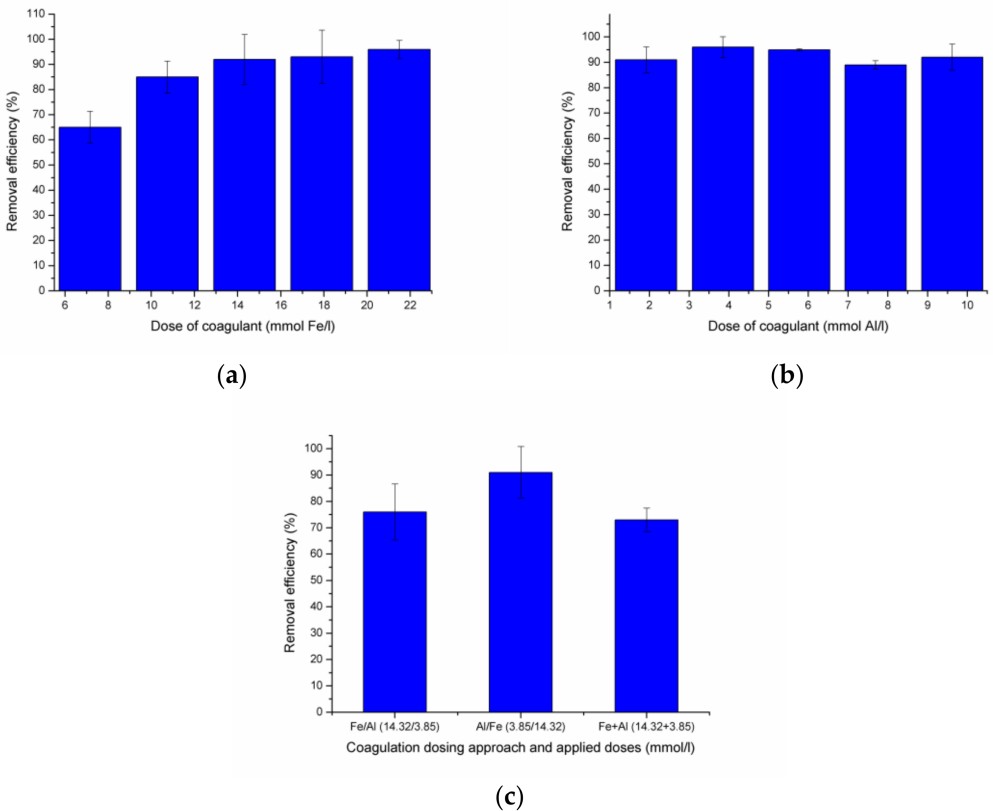

**Figure 10.** Efficiency of removal of textile fibers from a laundry wastewater by coagulation with (**a**) FeCl₃, (**b**) PACl, and (**c**) a combination of FeCl₃ and PACl (error bars show standard deviation of duplicates).

The results presented in Figure 10 show that as the FeCl₃ dosage increases, the removal efficiency also increases, with the highest applied doses completely removing the textile fibers from the wastewater. Doses higher than 9.62 mM were also used, but were not found to be effective. This can be explained by the fact that at a high coagulant dosage, saturation occurs where the particles no longer form flocs and therefore sink to the bottom along with the textile fibers [18].

PACl was also shown to be an effective coagulant in the removal of textile fibers from laundry effluents, as evidenced by the fact that the lowest removal efficiency was as high as 89% compared to initial values. All coagulant concentrations achieved similar removal efficiencies. When both coagulants were used in combination, the values were found to be slightly lower, but still considered effective for the removal of textile fibers.

The lower removal rates obtained in laundry wastewater compared to those in surface water can be explained by the presence of surfactants in this mixture. This is in agreement

with the results of Li et al. (2022) [47], who demonstrated a low efficiency of FeCl$_3$ in removing MPFs from surfactant-rich laundry wastewater. Moreover, these studies indicated the lower effect of surfactants on PACl performance during coagulation compared to FeCl$_3$, which is also confirmed by our results (Figure 10b).

Using a combination of coagulants, for both surfaces and wastewater, the best combination was when PACl was added first followed by FeCl$_3$, with removal rates of 86% for surfaces and 91% for wastewater. For the synthetic matrix, the best result was obtained with the simultaneous addition of both coagulants, with a removal rate of 78% compared to the initial concentration.

According to the literature data, the removal efficiency depends on the type of material to be removed, i.e., its properties. From the literature data [25,48], it can be concluded that microplastics, such as polyethylene, polystyrene, and polypropylene are removed more efficiently by using PACl in deionized water and drinking water, while the results of this study show a better removal rate when FeCl$_3$ was applied in similar water matrices. Moreover, according to the literature [24,49–51], it is clear that the effectiveness of the coagulant also depends on the characteristics of the water matrix, which was also confirmed by this work, as much higher coagulant dosages were required to effectively remove textile fibers from laundry wastewater compared to surface water and synthetic matrix. The type and dosage of coagulant is another factor affecting the treatment efficiency, where PACl was more effective on laundry wastewater, while Fe-based coagulants were more effective at removing textile fibers from synthetic matrix and surface water. Therefore, future studies should focus on optimizing the coagulation–flocculation treatment (e.g., by adding activated carbon) to reduce the coagulant dose required for efficient treatment and better removal of textile fibers from water.

## 4. Conclusions

As a contribution to a better understanding of the behavior of this type of pollution in wastewater treatment, the coagulation/flocculation treatment of different types of water containing a complex mixture of textile fibers was studied. It was found that a mixture of textile fibers containing natural and synthetic microfibers can be effectively removed from different water matrices with different compositions by using FeCl$_3$ and PACl as coagulants and their combination under near neutral pH conditions. The results show that high dosages of coagulants (>0.7 mmol/L for synthetic and surface waters; >7 mmol Fe/L and >1.9 mmol Al/L for laundry wastewater) are required to achieve efficient removal of textile fibers. Natural organic matter enhance textile fiber removal during coagulation/flocculation treatment. The highest removal efficiency of both coagulants was demonstrated in surface water, where complete removal was achieved by the application of FeCl$_3$ and almost complete removal was achieved by the application of PACl. Surfactants can reduce the efficiency of coagulation/flocculation in removing textile fibers from water when FeCl$_3$ is used as a coagulant, while the efficiency of PACl is actually increased. Natural fibers make it difficult to identify synthetic fibers in the mixture, which can lead to an underestimation of the amount present in the water, and thus the treatment efficiency.

**Supplementary Materials:** The following supporting information can be downloaded at: https://www.mdpi.com/article/10.3390/pr11030820/s1, Table S1: Doses of coagulants used in experiments; Table S2: Concentration of textile fibers in synthetic water after coagulation/flocculation; Table S3: Concentration of textile fibers in surface water after coagulation/flocculation; Table S4: Concentration of textile fibers in laundry wastewater after coagulation/flocculation.

**Author Contributions:** Conceptualization, J.A.; methodology, M.V., J.A., S.F., A.T.; formal analysis, S.F., R.T.; investigation, S.V., M.V., S.F., S.D.; resources, J.A.; data curation, S.V., R.T.; writing—original draft preparation, S.V.; writing—review and editing, M.V., J.A., S.F., S.D., A.T.; visualization, S.V., S.D.; supervision, A.T.; project administration, A.T. All authors have read and agreed to the published version of the manuscript.

**Funding:** Provincial Secretariat for Higher Education and Scientific Research: 142-451-3182/2022-01/2; European Cooperation in Science and Technology: CA20101.

**Data Availability Statement:** The data presented in this study are available on request from the corresponding author.

**Acknowledgments:** The authors gratefully acknowledge the support of the Provincial Secretariat for Higher Education and Scientific Research (Project No. 142-451-3182/2022-01/2). This article is based upon work from COST Action Plastics monitoRIng detectiOn RemedIaTion recovery—PRIORITY, CA20101, supported by COST (European Cooperation in Science and Technology).

**Conflicts of Interest:** The authors declare no conflict of interest.

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
