# Peer review of "Efficiency of Coagulation/Flocculation for the Removal of Complex Mixture of Textile Fibers from Water"

_processes, doi:10.3390/pr11030820_

Round 1

Reviewer 1 Report

1.The abstract must be rewritten. Just use the important goals and findings.

2. The quality of figures must be improved.  In case of figure 1 the font inside the figure must be corrected.

3. explain more about the size of fibers in SEM part.

4.Conclusion part must be refined. the present form must be shortened.

Reviewer 2 Report

Review of processes-2222281 “Efficiency of coagulation/flocculation for the removal of complex mixture of textile fibers from water” Sanja Vasiljević, Maja Vujić, Jasmina Agbaba, Stefania Federici, Serena Ducoli, Radivoj Tomić and Aleksandra Tubić

This study concentrates on coagulation/flocculation process efficiency in removal of a mixture of textile fibers from various water matrices (synthetic matrix, surface water and laundry wastewater), using either ferric chloride, polyaluminium chloride or a combination of the two agents.

The manuscript content and scientific development is reasonably good.

Some additional comments are included below:

1.     The pH influence mentioned in abstract has not been investigated, even though it is a parameter of outstanding significance in this case. If no other information can be added at this point, please mention what was the pH the measurements were performed at?

2.     Please include comparisons with other available literature data for these agents ( FeCl3 and PAlCl), for different types of water.

3.     Parameters influencing the flocculation/coagulation processes such as temperature and effluent quality should be discussed and some values should be reported within the manuscript. Based on three different water types, could you debate how they impact the removal efficiency.

4.     Minor English corrections are necessary.

Reviewer 3 Report

The authors were to evaluate of the removal efficiency of textile microfibers in three types of wastewater simulating the scenarios in wastewater in laundries. This manuscript is not novel, since the flocculation/coagulation process with FeCl3 and PACl are already widely studied and known in the removal of textile fibers. 

for example, Effect of coagulation on microfibers in laundry wastewater https://doi.org/10.1016/j.envres.2022.113401 (to mention one)

Together with the above, it presents a deficient methodological description, it does not present an appropriate experimental design, the analyzes were carried out in duplicate instead of triplicate, and the discussion of the results is very poor.

Round 2

Reviewer 3 Report

Dear Authors, I hereby inform you that enriched the manuscript adequately, improving the discussion and clarifying the innovation of the work, the version is appropriate for publication in the journal. However, he left some important observations that you must make before the manuscript is suitable for publication.

1. Please review all chemical formulas and subscript the number of atoms. Check ferric chloride formula lines 21, 31, 32, 317.

2. The temperature unit should not have a gap.

3. In table 1 there is a mistake in the temperature unit, the degree is after C and must go before.

4. Please, in the characterization, specify which were volumetric techniques used to determine (COD), chlorides, sulfates and bicarbonates, it is very important.

5. It is important to mention the methodology used for the determination of metal ions such as Na, Mg, K, and Ca, since they are in the results but do not describe how they were obtained.

6. Please explain what you mean by “*method detection limit”

7. On line 219 please separate “differencesin”
